# Interventions across the Retirement Transition for Improving Well-Being: A Scoping Review

**DOI:** 10.3390/ijerph17124341

**Published:** 2020-06-17

**Authors:** Miriam Rodríguez-Monforte, Carles Fernández-Jané, Anna Martin-Arribas, Lluís Costa-Tutusaus, Mercè Sitjà-Rabert, Inés Ramírez-García, Olga Canet Vélez, Jenna Kopp, Jordi Vilaró, Elena Carrillo-Alvarez

**Affiliations:** 1Global Research on Well-Being (GRoW), Blanquerna School of Health Sciences, Universitat Ramon Llull, Padilla, 326-332, 08025 Barcelona, Spain; miriamrm@blanquerna.url.edu (M.R.-M.); lluisct@blanquerna.url.edu (L.C.-T.); mercesr@blanquerna.url.edu (M.S.-R.); tomeshjen@gmail.com (J.K.); jordivc@blanquerna.url.edu (J.V.); elenaca@blanquerna.url.edu (E.C.-A.); 2GHenderS Research Group, Blanquerna School of Health Sciences, Universitat Ramon Llull, Padilla, 326-332, 08025 Barcelona, Spain; annama7@blanquerna.url.edu (A.M.-A.); inesrg@blanquerna.url.edu (I.R.-G.); olgacv@blanquerna.url.edu (O.C.V.)

**Keywords:** retirement, transition, intervention, well-being, scoping review

## Abstract

(1) Background: The work-to-retirement transition involves a process of psychologically and behaviorally distancing oneself from the workforce that is often accompanied by other social changes, which can influence health and well-being. However, research on interventions targeting the work-to-retirement transition to improve health status is limited. Our objective was to summarize and describe interventions aiming to improve well-being across the retirement transition; (2) Methods: We conducted a scoping review following the methodological framework described by Arksey and O’Malley; the Joanna Briggs Institute guidelines as well as the PRISMA-ScR statements; we systematically searched articles and gray literature to identify interventions and policies that aimed to improve well-being across the retirement transition. (3) Results: 15 publications were included, which comprised both experimental designs (*n* = 10) and systematic reviews (*n* = 5). (4) Conclusions: More research on how to promote overall well-being during the work-to-retirement transition is needed. The results of this scoping review show that most reported interventions address one single lifestyle behavior, and that relevant social determinants of health have been barely considered in their design. Future investigations need to consider vulnerable groups and country-specific structural conditions. Adopting a patient and public involvement approach will contribute to developing interventions that address the significant needs of those in the transition to retirement.

## 1. Introduction

The work-to-retirement transition involves a process of psychologically and behaviorally distancing oneself from the workforce and is often accompanied by other social changes in daily routine, the involvement in social groups, or income acquisition that can entail lifestyle modifications. The person faces new social roles, expectations, challenges, and opportunities which can impact lifestyle and health status [1,2,3,4,5,6,7]. Observational research has shown significant changes in lifestyle indicators such as physical activity, sedentary behavior, dietary habits, or socialization [8,9,10], and also regarding mental health [2,3,11]. However, precisely because transitions bring along normative lifestyle changes, synchronizing intervention programs with life transitions has been proposed as a potential opportunity to increase effectiveness [12,13], although research on the transition to retirement has been found to be scarce. Specifically, more research on how individuals experience and face this transition, and what factors can interfere or contribute to the promotion of healthy lifestyles has been asked for [1,2,3,4,5,6,7,8,9,10,11].

Interventions to promote health and well-being during the transition to retirement can be crucial, not only to foster adjustment in this moment of the lifespan but also because of their potential contribution to support healthy aging and to prevent frailty, both pressing issues worldwide and particularly in developed countries [14,15,16]. We specify the term well-being to imply that health promotion should not restrict itself to the biological nor physical aspects of health, but rather encompass the psychological, social, and transcendental dimensions of the person [17,18].

Increasing reports and reviews of interventions aimed at improving health and well-being in people at retirement age have been published recently [19,20,21,22,23]. However, the impact of different interventions during retirement on lifestyle behaviors and health outcomes is still unknown. From the perspective of the theories of the social determinants of health - that posits that the conditions in which people are born, grow, live, work, and age heavily impact health behaviors and health outcomes [24,25,26], and of the accumulation of risk and the life course approach—which emphasizes a temporal and social perspective, according to which that the cumulative effect of such living and working conditions are risk factors as one ages [27,28,29], relevant questions to be clarified comprise a better understanding on how interventions can be tailored to the living and work conditions of the retirees, especially in terms of length, behavioral areas to be tackled, delivery mode, and segmentation.

A clearer knowledge of interventions that can support a positive lifestyle during the work-to-retirement transition could contribute to foster healthy aging and prevent frailty in aging populations and is important for healthcare providers, social stakeholders, and policy-makers to ensure that they are adequately assisting individuals in this transition. Therefore, we intend to conduct a scoping review with the aim of describing interventions to improve well-being across the retirement transition in a biopsychosocial way. Our research questions include: Which interventions exist in order to improve the well-being of adults around the transition age? What is the current evidence of these interventions for improving the well-being of this population? Which settings and professionals are involved in conducting these interventions?

## 2. Materials and Methods

The protocol with the scoping review methodology has been published, but we will briefly describe it below. We considered the methodological framework described by Arksey and O’Malley with improvements by Levac, Daudt, and Colquhoun. We also followed the Guidelines published by the Joanna Briggs Institute as well as the Preferred Reporting Items for Scoping Reviews (PRISMA-ScR) [30,31,32]. The project was registered at Open Science Framework with the name “TRANSITS: work to retirement transition project” and published in the ninth volume BMJ Open [33].

### 2.1. Search Strategy

Two members of our team with expertise in review methodology developed a highly sensitive search strategy combining the following terms: retirement, health, policy, well-being, quality of life, clinical trial, RCT (complete search strategy for each database can be found in Online Appendix A). Medical and social science bibliographic databases screened were: Cochrane Central Register of Controlled Trials (CENTRAL), Medline, CINAHL, PsycINFO, and ISI Web of Knowledge. Reference lists of the included studies were also considered.

For gray literature, trial registries (Clinicaltrials.gov, EU-CTR), Google, Google Scholar, Yahoo, and Opengrey were used. Specific and recommended resources for searching gray literature were followed [34].

### 2.2. Eligibility Criteria

To be included in the review, the following eligibility criteria were considered: (a) inclusion of all type of original studies (quantitative, qualitative, mix-method studies), review articles or reports published on journals as well as gray literature; (b) describing interventions to improve the well-being in adults across their retirement transition; (c) including participants before, during and after retirement; (d) all publications must describe variables associated with participants’ physical well-being and/or psychological well-being and/or social well-being and/or their perceived quality of life related to these; (e) with no language restriction and (f) covering the time frame to April 2020.

### 2.3. Screening and Study Selection

All documents were screened by two independent investigators, first based on title and abstract and second using the full text papers or documents. Discrepancies were resolved by consensus.

Authors of the original studies were contacted if relevant information on eligibility or key study data was not available in the published report (Figure 1).

### 2.4. Data Abstraction

Data were abstracted by two independent reviewers and compared. A standardized form was created by the research team in order to collect the data (Microsoft Excel Spreadsheets). We followed the Joanna Briggs Institute reviewer’s manual 2015 for data charting [31].

The data abstracted included the following items:Author, year of publication, journal, or other information source.Study population characteristics (ethnicity, age, sex, educational level, presence of physical, psychological, or social problems at baseline).Transition to retirement definition.Design of the study.Follow-up and retention rates.Intervention (type, setting, professional/person involved in the delivery, duration).Outcome effects (including measurement approach and tools).Outcome analysis.Study quality.

### 2.5. Quality Assessment

Despite being considered an optional step, the team agreed on assessing the quality of the included studies in our review. We used the mixed methods appraisal tool (MMAT) for the assessment of the RCT’s as well as a tool for assessing the methodological quality of systematic reviews (AMSTAR) for the systematic reviews. Both of the tools have been validated and allow the inclusion of the different types of studies included in our review [35,36].

### 2.6. Data Analysis

The main findings were summarized using a narrative descriptive synthesis approach and grouped following the PCC principles to link the different findings to the review question/s.

### 2.7. Consultation

We organized a meeting comprising 21 stakeholders which included a variety of profiles based on their experience or expertise on the topic: recently retired persons, persons about to retire, community nurses, physicians, social workers and psychologists, Catalonian Health Department representatives, and retiree associations’ leaders. The main objective of the meeting was to obtain feedback on the research findings and to develop the next steps in research and practice. The feedback from the stakeholder meeting and the results of the scoping review were combined to clearly indicate the available evidence, gaps in research, and future research priorities for this population.

## 3. Results

We initially identified 1031 records. After removing 347 duplicates, the remaining 684 records were screened for title and abstract. Afterwards, the remaining 46 records were assessed for eligibility using full text. Finally, 15 publications were included (Figure 1), which comprised both experimental designs (*n* = 10) and systematic reviews (*n* = 5). For clarity purposes, we report the results of these study designs separately.

### 3.1. Experimental Designs

Details of the study participants and general characteristics of the experimental studies are summarized in Table 1 and Table 2.

#### 3.1.1. General Characteristics of Included Studies

##### Study Design and Country

The reviewed experimental studies were performed in Canada [37,40,41,42], the USA [38,39], the Netherlands [23], the UK [22], Belgium [44] and Australia [43]. The designs comprised two quasi experimental trials [40,41], three pilot RCTs [22,42,43] and five RCTs [23,37,38,39,44].

##### Participants

The number of participants in the overall trials accounted for more than 15,000. The participants ranged from less than 25 [41,42], to more than 10,000 [38,39].

With regard to the age of the participants, most participants’ mean age ranged from 55 to 65 years old. The mean age of the study participants by Fries [38,39], was slightly above this trend (68 years old).

Most studies reported gender characteristics and included both men and women, except for Cunningham [37], which included only men and Ashe [42] including only women. Gender composition of the sample was not reported by Werkman et al. [23].

Socioeconomic characteristics were not described in most of the included studies. Income level was reported by Cunningham et al. (1997) and ranged from less than 10.000 to more than 39.999 dollars/year [37]. Education level was reported in five studies [23,40,41,42,44]. Ethnicity/immigration status was reported in one study (Caucasian participants [41]).

Additional inclusion criteria regarding baseline conditions comprised diverse health characteristics including physical and psychological aspects: not having any particular health condition [40], not undergoing any medical treatment that might affect body composition [23], absence of depression or severe mental health conditions [22], being healthy inactive [42], being able to walk 100 m without assistance [44], presenting intellectual disability [43], or presenting suicidal ideations [41]. The rest of the included studies did not report any baseline characteristics inclusion criteria [37,38,39].

#### 3.1.2. Objectives and Retirement Definition

How “retirement” was conceptualized as an inclusion criterion was diverse among studies. Three studies included participants according to their age. Of those, one included participants that were 50 to 65 years old [40], and the rest were 55 to 70 or 75 years old [38,42]. Only one study included participants 45 years or older [43]. Three other studies used time before or after retirement, with a period ranging from two years before retirement to two years after retirement (being retired between 6 months and 5 years [44]; two years before or after retirement [22]; and 2 to 4 months prior to retirement [37]). Lastly, two studies combined age and time to/from retirement, including participants from 50/55 to 65 years old and retired for less than 6 months [23] or 6 years [41].

#### 3.1.3. Main Findings and Conclusions

The interventions described in the different studies included three main aspects: physical activity, healthy and dietary habits, and psychological and social support. The assessed interventions lasted between 5 weeks and 2.5 years (5 to 8 weeks [22,44], 6 months [42,43], 1 year [23,37], 2 years [38,39,41]). Concretely, the different interventions described in the studies were based on physical activity [37,42,44], social or/and psychological support [40,41,43], health education (Fries 93, 94) or a combination of the previous [22,23]. The intervention settings were, mainly, on-line or post based [22,38,44], or community group or home-based [23,37,40,41,42,43].

The diverse specific aspects and results of the interventions will be described separately based on their main objective in order to achieve homogenization and better abstract their general characteristics.

##### Physical Activity

Three trials investigated the effect of a physical activity intervention [37,42,44].

Ashe et al., conducted a physical activity intervention aiming at comparing an enhancement in physical activity, engagement, and satisfaction of the participants following tailored group-based physical activity plus education and social support strategies compared to the control group which only received education sessions; both groups participated for 6 months. Ashe reported that the participants in the intervention group increased their physical activity and reduced weight and blood pressure. High levels of engagement and satisfaction were also reported. The intervention was provided by personal trainers, a psychologist, and dietitians.

Van Dyck et al. conducted a 5-week self-regulation eHealth intervention in order to increase physical activity in recently retired adults. Authors reported short term (1 month) improvements in transport-related walking, leisure-time walking, leisure-time vigorous physical activity, moderate-intensity gardening, and voluntary work-related vigorous physical activity. The intervention was delivered through an online platform.

Cunningham and colleagues described a one-year exercise training aiming at assessing cardiorespiratory fitness, levels of daily leisure physical activity, and blood lipids. Results of the intervention group compared to controls showed an improvement in respiratory parameters (VO_2_, ventilation), but no significant results were found on blood lipids. The intervention was provided by an exercise leader.

##### Psychological and Social Interventions

Three studies aimed at improving psychological and social aspects from the participants [40,41,43]. Dubé and colleagues, conceptualized an intervention assessing the effectiveness of 10 h of goal-oriented group meetings seeking the management of personal aims, and the improvement of psychological and mental health among the participants. The indicators showed maintained improvements six months after the end of the study. The intervention was delivered by a retiree and a psychology graduate student.

Lapierre et al., also described an approach based on a personal-goal intervention in order to achieve well-being. The participants of this intervention had baseline suicidal ideations; therefore levels of depression were also assessed. The intervention included 10 to 12-h small group meetings conducted by a trained retiree and a psychology graduate student which were focused on helping the participants to achieve one meaningful personal project. Results showed significant achievements in terms of different psychological indicators (e.g., levels of psychological distress and depression decreased).

Stancliffe et al. designed an intervention which involved supporting older adults with disability to attend a mainstream community. Several psychological and social indicators such as loneliness, social satisfaction, depression, quality of life or social contacts, among others, were evaluated throughout the intervention which consisted of delivering active mentoring and support from community group members during 6 months, on a weekly basis. The intervention was delivered by non-professional trained mentors from the community.

##### Multimodal Interventions: Physical Activity, Dietary Habits, and Social Aspects

Two trials combined interventions of physical activity with other elements such as dietary habits, or social aspects [22,23].

Lara described a web-based intervention which comprised diverse modules, working aspects such as physical activity, healthy eating, and social connections. Results showed that participants visited the web on different occasions, and were mainly interested in moving more, eating well, and participating in social programs, finding the topics of physical activity and diet to be acceptable for the retirement transition. Werkman et al. designed a one-year low-intensity computer-tailored program and feedback in order to assess its impact on waist circumference, body weight and composition, blood pressure, physical activity, and dietary intake. Results, despite being non-significant, showed a pattern of small effects on body composition, physical activity, and dietary behavior. The providers of both interventions were not reported.

##### Healthy Habits

Two trials from the same authors (Fries et al.) evaluated health education emphasizing self-care [38,39]. Authors used individualized reports and recommendations based on health assessment and provision of educational materials compared to the control group which only received passive portions of the program. Authors concluded that after 12 months, health risk scores improved. Furthermore, there was a significant reduction in the utilization of medical services and, therefore, on health costs. The providers of both interventions were not reported.

#### 3.1.4. Quality Assessment

The quality of the included studies was analyzed following the criteria from the MMAT tool. In general, studies fulfilled most of the quality criteria reported by MMAT. However, only one of the studies reported blinding the outcome assessors of the intervention [22]. Three studies did not report how randomization was performed [37,41,43].

### 3.2. Systematic Reviews

Details of the included systematic reviews are summarized in Table 3.

#### 3.2.1. General Characteristics of Included Studies

The systematic reviews (SR) analyzed in our study were conducted in the UK [20,46], Portugal [47], and Canada [45,48]. The SR published by Baxter, Loureiro, Heaven, and Wilson specifically analyzed the transition to retirement period, while the one published by Vrkljan analyzed several life transitions, including transition to retirement [45].

Studies analyzed in each SR ranged from 0 [47] to 64 [46]. Heaven et al. included 7 different interventions, whereas Wilson et al. (2007) reviewed 20 observational studies. Vrkjlan et al. (2018) assessed 2 studies that evaluated interventions to support the transition to retirement.

Participants sociodemographics’ from the included studies in the different SR described diverse age ranges. Most trials included participants from 55 to 75 years old, with one broadening the range from 45 to 80 [48]. In terms of gender, some of the studies included within the SR did not report data on this aspect [47,48]. In the case of Baxter (2016), 5 studies did not report gender characteristics, whereas 11 studies included only the female population, which also accounted for a greater proportion in most studies. According to the rest of SR [20,45], gender was accounted for in every single study and included more proportions of women. From the 2 studies included by Vrilkjan et al., one of them only included men.

#### 3.2.2. Objectives and Retirement Definition

The SR had different objectives. Briefly, they aimed at synthesizing evidence on interventions to increase physical activity [46], promote positive social roles [20], support major life transitions [45], and promote overall health [47,48].

The definition of retirement varied across studies. In this way, Baxter et al., included studies where participants were going through the retirement transition or shortly after, including those not in paid employment, and those about to leave paid employment. In the absence of literature referring to their target population, they used age (older adults) as a proxy for the period of retirement transition. Heaven et al., considered participants to be in the retirement transition if their mean or median age was between 55 and 70 years old; if they were about to retire or had retired within the past two years; or if they had a partner who met one of these criteria. For Loureiro et al., participants included were those within five years of retirement. According to Vrkljan and colleagues, the definition included those studies reporting retirement from paid employment. Finally, Wilson and Pahla did not report any definition of retirement used as inclusion criteria.

#### 3.2.3. Main Findings and Conclusions

Loureiro et al. did not find any trial specifically conducted during adjustment to retirement, (within five years of retirement) and, therefore, did not identify any intervention.

Wilson et al., although aiming to evaluate methods for health promotion and describe long and short-term benefits, only included observational studies. Four main considerations were described after the analysis of their findings: the need for support in order to achieve more positive retirements, the need for identifying and overcoming barriers on health promotion in this life stage, considering the sustainability of health changes by assessing the methods to introduce health promotion and describing short and long term benefits of health promotion at retirement.

Vrkljan et al. aimed at examining the effectiveness of interventions targeting several life transitions. Authors included two trials that focused on the transition to retirement. The included interventions (5 weeks follow-up) were based on group meetings focused on expressing feelings related to thankfulness for one’s family, forgiveness for previous wounds inflicted, and love for one another to maintain harmonious relationships and peaceful living, and workshops to identify, explore and clarify strengths and passions. The authors suggest that, although considering that more studies are needed, group-based approaches provided by trained professionals might be positive for the adjustment/transition to retirement, spouse and resulting in lower oxidative stress levels (main outcomes of the SR).

Heaven’s review identified 11 studies about social roles, including follow-ups ranging from 6 months to 24 months. Social roles identified encompassed: acting as “grandparents” to neglected children, assisting in local schools and kindergartens, undertaking placements in local voluntary organizations, acting as mentors for newer workers in the organization in which the participants were employed and conducting gardening and maintenance work in local parks. Intergenerational contact was central to three of the interventions. The outcomes measured different concepts such as life satisfaction, perception of age and aging, productivity and self-actualization, social support and social activity, functional health, cognition, and mental and psychological health. The conclusions drawn by the authors consider that interventions offering an explicit social role with group support can improve health and well-being for those who carry out the role. However, major sources of bias affecting the studies in the review limit how confidently the interventions were effective.

Baxter’s described interventions (6 weeks to 4 years follow-ups) including counseling and advice, group sessions, individual exercise programs, in-home telephone interventions, in-home combined diet and exercise interventions, home-based interventions providing pedometer/accelerometer, computer-based interventions, and community-wide initiatives. Outcomes included a set of physical measures such as blood pressure, body mass index, cardiovascular risk assessment, physical activity questionnaire, etc. Despite the high number of studies included in their review and a majority of studies reporting some intervention effect, with evidence of positive outcomes for all types of programs, they were unable to find any evidence that the transition to retirement period was, or was not, a significant point for intervention. Studies in older adults more generally indicated that a range of interventions might be effective for people around retirement age.

#### 3.2.4. Quality Assessment

The quality of the included studies was analyzed following the criteria of the AMSTAR tool. In general, studies did not fulfill some of the quality criteria reported by AMSTAR. Most of the studies did not report or partially reported a comprehensive literature search strategy. Moreover, the concretion of following a duplicate study selection and extraction process was present in one study [20]. The assessment of their sources of bias [20,46] as well as the discussion of the sources of heterogeneity [20,45] were only described in some studies.

### 3.3. Gray Literature: Collection of Observational Data on Retirement and Healthy Aging

The search conducted in order to synthetize the available data on interventions in the work-to-retirement transition did not retrieve any findings. Therefore, we have synthetised the available observational data on this topic which could serve as a platform in order to design and develop future interventions during this transition period.

Some countries across the world are collecting observational data on aging and retirement in order to assess and improve healthy aging, where the retirement transition is the starting point. The Gateway to Global Aging Data includes a digital library of survey questions, a search for finding comparable questions across surveys, and identically variables for cross-country analysis [49]. In Europe, the SHARE multidisciplinary and cross-national panel database includes micro-data on health, socio-economic status and social and family networks of about 140,000 individuals aged 50 or older, covering 27 European countries and Israel [50]. The WHO developed the longitudinal study on global aging and adult health which collects data on adults aged 50 years and older, plus a smaller comparison sample of adults aged 18–49 years, from nationally representative samples in China, Ghana, India, Mexico, Russian Federation and South Africa [51]. Other countries such as Brazil, Costa Rica, Mexico, South Africa, China, India, Indonesia, Malaysia, Thailand, Japan, Korea, England, Ireland, Northern Ireland, Scotland, and the USA have also collected longitudinal data on aging and retirement [52].

Countries such as Japan or Spain with one of the highest life expectancies worldwide are focusing on the collection of data on adults close to their retirement (50 years or older). The sustainability of the health and social system relies on the quantification of these populations and the subsequent development of preventive and health promotion strategies aimed at healthy aging [53,54].

### 3.4. Stakeholder Consultation

Twenty-one stakeholders including recent retirees or professionals about to retire, health and social professionals, and policymakers at different competence levels attended our consultation meeting in person. The session was structured in three different parts. During the first part of the session, we presented the results of our review with a final note that included the formulation of questions in regards to the main findings of our review: (1) how would they define “transition to retirement”; (2) what physical psychological and social needs emerge during the transition to retirement and whether these change based on work and life conditions; (3) what resources already exist to support people during this transition; (4) how would ideal resources for this transition would like; (5) when should these resources be provided to optimize their impact and sustainability. The second part included small group discussions to answer questions presented in part one and provide additional comments to be shared with the rest of the group.

The main ideas developed during the session comprised: the importance of the social dimension and physical activity as core concepts for the dynamization of the retirement transition. A good and sustained social network accounts for a better transition to retirement as well as previous preparation and adaptation during other life transitions. Women usually adapt better to life changes and, in the case of the retirement transition, tend to adapt better than men for their capacity to diversify on the activities conducted during their life. Activities during the retirement transition should be flexible and adapted to the person’s choices and needs. The same ideas apply in the preparation stage and during the transition to retirement. There are differences in the type of needs, structures, and activities in urban and rural areas. Intergenerational activities and spaces are important, as well as the development of the figure of the mentor for younger persons. The mentor figure was also considered a good resource in terms of facing the retirement transition being supported by someone who has experienced the transition before. To close the session, participants expressed the need to conduct more in-depth research in order to hear the voices of people with a different profile or vulnerable populations.

## 4. Discussion

To the best of our knowledge, this is the first scoping review to collate and analyze interventions to promote well-being across the retirement transition. Following current guidelines for scoping review conducting, peer-reviewed papers, gray literature, and a stakeholder consultation have been included with the aim of contributing in a holistic manner to the knowledge on how to promote well-being across the retirement transition. Specifically, we set three research questions: (1) Which interventions exist in order to improve the well-being of adults around the transition to retirement? (2) What is the current evidence of these interventions for improving the well-being of this population? (3) Which settings and professionals are involved in conducting these interventions? These will be addressed in this discussion in order to conclude with implications for both research and practice.

When answering the first research question, the most critical point of this review stems from the conceptual vagueness of well-being and transition to retirement as core notions of this research. In this way, only two of the ten experimental studies [22,23] and no systematic review addressed more than one dimension of the ones in the bio-psycho-social model considered by the WHO in its definition of health [18] or the health and well-being continuum described by Kirsten et al. (2009)—which served as a theoretical basis for this research [17]. Moreover, most evidence was produced in countries such as Canada, the USA, the UK, Belgium, and Australia, and, as such, it reflects the cultural and socioeconomic reality of those countries, therefore not necessarily being applicable to other contexts, not only in terms of culture and living conditions, but also with regard to retirement terms—as shown in the last OECD report on pension markets [55]. Moreover, when conducting our search in order to identify gray literature on the topic, no results were found. The lack of interventions on this specific transition period must be considered and improved, due to the important impact that primary prevention of diseases and syndromes such as frailty might have on the quality of life of these populations. However, there exists a considerable amount of observational data that might be the basis in order to design and develop future interventions.

The heterogeneity of operationalizations of the notion “transition to retirement” found in our sample of studies may constitute an added barrier towards making research pertinent to people across the retirement transition. First, because research results are not comparable nor easily applicable. Second, because using age as a proxy for being in the transition to retirement, especially when wide ranges are included (45–75 years old, in our sample), may end up in samples with individuals that are not necessarily in the transition to retirement, thus biasing the results. In fact, according to the contributions from the participants in the sixth phase of this scoping review, the most relevant time-period in terms of adjustment to retirement is rather narrow, ranging from 6 months before to 2 years after retirement and not necessarily related to age.

Additionally, the fact that the social determinants of health are not adequately taken into account, as shown by the observation that most researches do not describe nor adapt to socioeconomic and demographic variables despite the evidence that observational studies provide on the impact they have on the retirement transition [8,56], limits the extent to which the results of this review can be extrapolated to various settings or applicable to different individual profiles, which goes against the health promotion basic premise of tailoring interventions to achieve better impact [57].

Hence, the response to the second question: What is the impact and effectiveness of these interventions on the present and future well-being of this population? is unclear, due to the high variation of interventions, contexts, participants, and deliveries. Moreover, despite not excluding the papers due to quality issues, the quality assessment of the papers included in our scoping review reveals that there exists significant room for improvement.

The last point of interest in our scoping review comprised the settings and professionals involved in conducting these interventions. A wide spectrum of both professionals and settings have been used, including psychologists, personal trainers, exercise physiologists, dietitians, or peer retirees—although many researches did not report the professional in charge of the intervention. The most used settings were community facilities, at-home delivery, and online resources, consistently with the current increase in e-health programs and initiatives [58,59]. If we consider the inputs from the consultation phase of this research, it would be desirable to foster delivery methods and settings that promote social interaction and help the recently retired person to dynamize his or her social sphere. Accordingly, results from some of the studies analyzed also account for significant improvements when these methodologies are applied, even in psychologically vulnerable individuals [40,41]. A further element of discussion is the fact that most studies reviewed did not integrate already available resources (from city councils, civil associations, etc.) nor consultation with those to be included in their research, thus jeopardizing its sustainability.

This review is not exempt from limitations. The specification of several inclusion criteria, particularly the requirement that is not age but retirement conditions what defines the participants in the selected studies (although in some cases retirement is defined through age as we have seen), may have resulted in the exclusion of studies that included individuals in their transition to retirement but that were not described as such. Our final sample comprised 15 articles (10 experimental studies and 5 systematic reviews) reporting interventions based on physical activity, dietary habits, health education, and social roles, most of which were informed positive results. However, it is not possible to establish practice recommendations based on these results due to heterogeneity and quality issues. In addition, during the stakeholder consultation, despite selecting a wide number of representatives on the topic, missing profiles such as retirees from vulnerable contexts might be a limitation that should be improved in future research.

On the other hand, the comprehensive systematic search of electronic databases and gray literature, the consultation phase, and a sound theoretical foundation are strengths of this review, which allow us to identify the state of the art in the topic of work-retirement transition and some recommendations for research.

As identified by previous studies in the field [7,8,9,10,11,12,13,14,15,16,17] studies on the retirement transition remain scarce and the vast majority use an observational design. At the same time, the results of the interventions included in this scoping review seem to indicate that interventions to promote well-being during this transition can yield positive results in terms of lifestyle changes and general adjustment to the new stage of life. However, the small number of studies, their heterogeneity, and their medium-low quality do not allow us to formulate strong conclusions. Along the same lines, current research does not permit to establish whether these interventions are useful to prevent frailty.

Future investigations on how to promote well-being across the retirement transition should have a clear rationale on how the retirement transition may be critical to health and well-being and adapt the intervention accordingly, particularly in terms of delimitation of the time-frame covered by the intervention, behaviors and health areas to be addressed by the intervention, and of segmentation to population subgroups based on living and working conditions prior to retirement. Contextual aspects such as the national and regional retirement system and political structures need to be taken into consideration, as key determinants of how the transition to retirement unfolds. In this sense, more research in South European countries is needed, to take into consideration its socioeconomic, labor, and cultural particularities. Special attention should be bestowed to individuals and groups in vulnerable situations because of informal working conditions or other circumstances. Observational research to clarify how these situations affect health and well-being during the transition to retirement is timely. Last, our results highlight the need for high-quality research based on more standardized criteria for data collection and the integration of patient and public involvement (PPI) approaches so that research is relevant to those it is addressed to.

## 5. Conclusions

Several interventions have been tested with regard to their contribution to promoting well-being in the transition to retirement. They include aspects of physical activity, social roles, health education, and psychological adjustment. However, as a result of this scoping review, we can state that more research on how to design and implement interventions to promote overall well-being during the work-to-retirement transition is needed since most reported interventions address one single lifestyle behavior, and that relevant social determinants of health have been barely considered in their design. Future investigations need to consider vulnerable groups and country-specific structural conditions. Adopting a PPI approach will contribute to developing interventions that address the significant needs of those in the transition to retirement.

## Figures and Tables

**Figure 1 ijerph-17-04341-f001:**
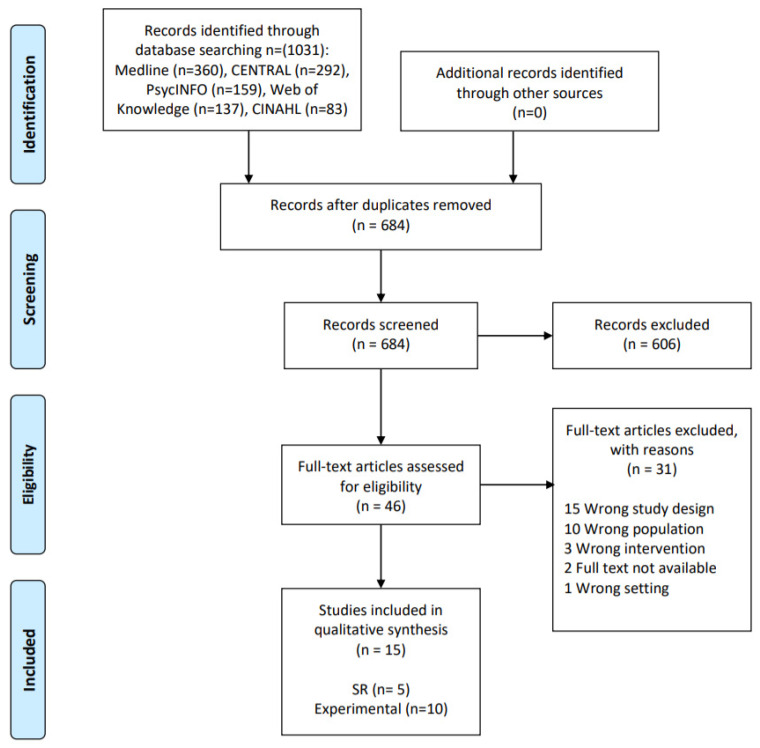
Flow diagram of study selection process.

**Table 1 ijerph-17-04341-t001:** Details of participants in the experimental designs.

Author, Year	Number of Participants (Male: Female)	Age: Mean (SD)	Ethnicity	Economic Level	Education Level	Presence of Physical, Psychological, or Social Problems at Baseline (Inclusion Criteria)
Cunningham, 1987 [37]	224 (224: 0)	62.7	NR	$10,000 (C: 11.7, I: 8.8) 10,000–19,999 (C: 46.8, I: 47.8) 20,000–29,999 (C: 27, I: 25.7) 30,000–39,999 (C: 9,0, I: 11.5) 39,999 (C: 5.4, I: 6.2)	NR	NR
Fries, 1993 [38]	1936 (977: 959)	I: 68.3 C: 68.7	NR	NR	NR	NR
Fries, 1994 [39]	12,838	63.6	NR	NR	NR	NR
Dubé, 2005 [40]	117 (81: 36)	I: 57.9 (3.26) C: 57.6 (3.01)	NR	NR	Mean years of education I: 14.1 (2.97); C: 14.2 (3.34)	None
Lapierre, 2007 [41]	21 (11: 10)	56.7 (1.9)	Caucasian	NR	Mean years of education: 15.2 (2.36)	Presence of suicidal ideations in the last week
Werkman, 2010 [23]	413 (352: 61)	I: 59.5 (2.0) C: 59.4 (2.0)	NR	NR	Low education level (primary school or low vocational education) I: 25%; C: 23%	Not undergoing medical treatments that might affect body composition
Ashe, 2015 [42]	20 (0: 20)	I: 64.8 (4.6) C: 63.1 (4.8)	NR	NR	Secondary education	Healthy inactive
Stancliffe, 2015 [43]	58 (32: 16)	I: 57.4 C: 53.8	NR	NR	NR	Presence of intellectual disability
Lara, 2016 [22]	75 (18: 57)	61.0 (4.0)	NR	NR	NR	Absence of depression or severe mental health conditions
Van Dyck, 2016 [44]	284 (134: 150)	I: 63.1 (2.2) C: 63.2 (2.0)	NR	NR	University degree I: 55.1%; C: 53%	Ability to walk 100 m without assistance

**Table 2 ijerph-17-04341-t002:** Measures and main findings of experimental studies.

Author, Year	Country (C) Settings (S) Funding (F) Design (D)	Aim	Transition to Retirement Definition	Intervention Description (ID), Provider (P) and Duration (DU)	Control	Outcomes Measures (OM), Endpoint (E)	Main Findings
Cunningham, 1987 [37]	C: Canada S: NR F: Ontario Ministry of Health D: RCT	To evaluate the effects of one-year exercise training on cardiorespiratory fitness, levels of daily leisure activity, and blood lipids (CT and HDL-CT)	2–4 months prior to retirement	ID: Physical Activity: three group sessions per week (plus one day on their own). Each training session consisted of a 10- to 15-min warm-up, 30 min of walking or jogging at an individualized pace, and a 10-min warm-down P: exercise leader DU: 1 year	No intervention	OM: anthropometry measures (weight, fatness), cardiorespiratory fitness (VO_2_, respiratory exchange ratio, heart rate; ECG, treadmill test), blood lipids (CT, HDL-CT), leisure activity levels (Minnesota Leisure Time Activity questionnaire), grip strength and flexibility E: 1-year post-intervention	VO_2_ increased compared to controls. There were no significant changes in maximal heart rate and respiratory exchange ratio although ventilation increased significantly in the trained group No significant differences were found between the groups with respect to the remaining outcomes
Fries, 1993 [38]	C: USA S: On-line/post mail F: Blue Shield of California, and Bank of America D: RCT	To evaluate the effectiveness of a health promotion program in a retiree population in terms of health risk and medical cost reduction	55–75 years	ID: health education emphasizing self-care: individualized reports and recommendations based on health assessment and provision of educational materials P: self-management DU: 24 months	Education regarding risk appraisals only, without feedback, for the first 12 months and subsequently the full intervention for the second 12 months	OM: health risk score (Framingham multiple risk logistic for CVD); health habits/indicators (seat belt use, BMI, dietary and saturated fat intake, cigarette-smoking, exercise) E: 6, 12, 18, 24 months	Health risk scores improved by 12% at 12 months compared with the control and by 23% at 24 months Individual health habit changes were favorable for all parameters studied and were highly statistically significant for most variables (except weight, CT and BP)
Fries, 1994 [39]	C: USA S: On-line/post mail F: Blue Shield of California, National Health Management Foundation D: RCT	To evaluate the cost trend reduction from a health promotion program	NR	ID: health education emphasizing self-care: individualized reports and recommendations based on health assessment and provision of educational materials P: self-management DU: 12 months	Passive portions of the program, including the self-management book and other educational materials.	OM: Health risk scores (Framingham multiple-risk logistic function); health habits/indicators (SBP, CT, seat belt use, pounds over ideal weight, high salt intake, salt dietary fat, cigarette smoking, alcohol intake, exercise, stress); self-reported medical utilization; claims paid E: 6, 12, 18 months	The program was associated with a reduction in health risk scores at 12 months, a reduction of subject reported medical utilization from baseline and a decrease in claims cost growth relative to controls
Dubé, 2005 [40]	C: Canada S: NR F: NR D: Pre-post	To evaluate the effectiveness of a goal-oriented intervention offered to retirees in order to increase their psychological well-being and their mental health and specifically to promote the expression, planning and realization of personal projects through a learning process based on a cognitive approach	50–65 years	ID: social support and psychological adjustment: management of personal aims. The program consisted on 10 to 12 meetings a total of 10 h in groups of 7 to 10 people P: retiree and a psychologist graduate student DU: the duration of the 10 to 12 meetings	No intervention	OM: the psychological well-being was assessed using five different indicators and instruments: emotional well-being during retirement (Retirement Experience Assessment scale; Short-term happiness (Short Happiness and Affect Research Protocol (SHARP); meaning of life (Ryffs’ six scales of psychological well-being; inner peace (the Serenity Scale); distress during the past weeks (index of psychological distress) E: at the end of the intervention and 6 moths post intervention	The experimental group improved significantly more than the control group on all the goal and subjective well-being indicators Gain was maintained six months later
Lapierre, 2007 [41]	C: Canada S: NR F: Conseil Québécois de la Recherche Sociale D: Quasi experimental	To evaluate a personal goal intervention program for early retirees in terms of their subjective well-being and levels of depression in the participants with suicidal ideations	50–65 years and being retired for more than 6 years	ID: psychological adjustment: cognitive-behavioral approach with 10 to 12 meetings of 2 h each week for small groups. P: retiree and a psychologist graduate student DU: 2.5 years	No intervention	OM: the psychological well-being was assessed using different indicators and instruments: positive experience with retirement, happiness, serenity, Ryffs’ well-being dimensions (personal growth, self-acceptance, positive relationships with others, environmental mastery, purpose in life, autonomy), flexibility, tenacity, depression, and distress. E: 12 weeks and 6 months	The experimental group had improved significantly more than the control group on hope, goal realization process, serenity, flexibility, and positive attitude toward retirement The levels of depression and psychological distress significantly decreased These gains were maintained 6 months later
Werkman, 2010 [23]	C: The Netherlands S: Community and domiciliary F: Netherlands Heart Foundation D: RCT	To investigate the effect of a one year low-intensity computer-tailored energy balance program	55–65 years and being retired for more than 6 months	ID: physical activity and diet: one-year multifaceted program including these factors using computer tailored feedback P: NR DU: 12 months	Newsletters with general information about the study, such as study progress, and information about art exhibitions and city trips for instance.	OM: waist circumference, body weight and body composition, BP, physical activity and dietary Intake E: 12- and 24-months follow-up	The program did not have a significant effect on any of the outcomes though it showed a pattern of small, non-significant effects on changes in body composition, physical activity and dietary behavior. Transition to occupational retirement and/or participation in research had a greater impact than the intervention program itself
Ashe, 2015 [42]	C: Canada S: NR F: Canadian Institutes of Health Research D: Pilot RCT	To test study feasibility for the Everyday Activity Supports You (EASY) model which seeks to encourage movement through daily activities and utilitarian walking	55–70 years	ID: group-based education and social support, individualized physical activity prescription and use of an activity monitor (Fitbit) P: personal trainer or exercise physiologist, dietitian D: 6 months	Monthly education sessions, but did not receive information on the importance of exercise or how to sustain an active lifestyle. They had no interactions with the exercise professionals nor did they receive Fitbit monitors.	OM: recruitment and retention rates, satisfaction, physical activity (average daily step count), MVPA (min/day), sedentary behavior (min/day, percentage/day), BP, BMI, social connectedness, self-rated health, self-efficacy, and intentions for physical activity E: 3 and 6 months	The model was feasible to deliver in a community setting to women at retirement age. Partici-pants were highly engaged in, and satisfied with, the EASY model The intervention was effective to increase physical activity and decrease weight and BP
Stancliffe, 2015 [43]	C: Australia S: Domiciliary F: NR D: Pilot RCT	To examine the feasibility of supporting older adults with disability to attend a mainstream community group; the types of mainstream community groups or volunteering groups that participants joined; the duration of participants’ weekly attendance at their group; and changes in outcomes experienced by participants	≥45 years	ID: social support and psychological adjustment: active mentoring - support from community group members. P: non-professional trained mentors from the community DU: 6 months	Paired working individuals who did not receive any intervention	OM: self-reported or proxy-reported: depression (GDS, Mini PAS-ADD); aloneness Loneliness (MWLQ); social satisfaction (MWLQ); loneliness (UCLA Loneliness Scale); quality of Life (SF-36); life events (Mini PAS-ADD); participation and social contacts (Weekly logs) E: 6 months	The community participation increased, making an average of four new social contacts and decreasing their work hours. Intervention participants were more socially satisfied post-intervention than comparison group members
Lara, 2016 [22]	C: UK S: On-line F: Lifelong Health and Well-being Cross-Council research initiative, UK Health Departments D: Pilot RCT	To assess the feasibility and acceptability of the intervention, the trial design, the procedures and the outcome measures of a web-based platform (Living, eating, activity and planning through retirement; LEAP) promoting healthy eating, physical activity and meaningful social roles	≤2 years of retirement, pre-retired or to be retired in the following 2 years	ID: physical activity, healthy eating, meaningful social connections. The intervention comprises the following five modules: (1) time, (2) changing work, (3) moving more, (4) being social and (5) eating well P: on-line platform DU: 8 weeks	Usual care: comprehensive health information service which encourages people to make healthy choices	OM: anthropometric measures (weight, BMI, waist circumference, body fat mass, fat free mas, total body water); dietary intake (multiple pass 24-hour); physical activity (accelerometer) feasibility and acceptability (completion rates and qualitative feedback) E: 8 weeks	“Moving more,” “eating well,” and “being social” were the most visited modules. At interview, participants reported that diet and physical activity modules were important and acceptable within the context of healthy aging
Van Dyck, 2016 [44]	C: Belgium S: On-line F: Research Foundation Flanders, Ministry of the Flemish Community, Department of Welfare, Public Health and Family D: RCT	To test the effectiveness of the self-regulation eHealth intervention ‘MyPlan1.0.’ to increase physical activity in recently retired Belgian adults	>6 months and <5 years of retirement	ID: *physical activity*: self-regulation eHealth intervention focused on pre- and post-intentional processes of behavioral change P: on-line platform DU: 5 weeks	No intervention	OM: Self-reported physical activity (IPAQ) E: 1 week and 1 month	At 1 week, the intervention significantly increased walking for transport. At 1 month, the intervention increased transport related walking, leisure time walking, leisure time vigorous physical activity, moderate intensity gardening, voluntary work-related vigorous physical activity

NR: Not reported; RCT: Randomized Control Trial; MVPA: Moderate to Vigorous Physical Activity; IPAQ: International Physical Activity Questionnaire; GDS: Glasgow Depression Scale; MWLQ: Modified Worker Loneliness Questionnaire; SHARP: Short Happiness and Affect Research Protocol; FLEX: Flexible goal adjustment; TEN: Tenacious goal pursuit; GRP: Goal Realization Process questionnaire; GIS: Goal Instability Scale; CVD: cardiovascular disease; ECG: electrocardiogram; CT: cholesterol; HDL-CT: high density lipoprotein-cholesterol; SBP: systolic blood pressure; BMI: body mass index.

**Table 3 ijerph-17-04341-t003:** General characteristics of systematic reviews.

Author, Year	Objective/Research Question	Inclusion Criteria				Main Findings
		Study Type	Participants	Intervention (I) Control (C)	Outcomes	
Vrkljan, 2018 [45]	To examine the effectiveness of published interventions across the three most common life transitions in older adulthood, namely, bereavement, retirement from paid employment, and relocation of residence to a higher level of care	RCT, non RCT and systematic reviews	Intervention targeting individuals in bereavement, retirement from paid employment, and relocation to a higher level of care	I: Any C: Any	Any	N = 2 Results suggest that group-based approaches provided by trained personnel can be effective, but further study is warranted.
Baxter, 2016 [46]	To synthesize international evidence on the types and effectiveness of interventions to increase physical activity among people around the time of retirement	Experimental and observational studies	People during and shortly after the transition to retirement In the absence of literature, age was used as a proxy for the period of retirement transition	I: Interventions which aimed to increase and/or maintain levels of physical activity. C: Any	Direct and indirect measures of physical activity, Social, psychological, behavioral and environmental outcomes.	N = 64 Little research has been conducted to assess whether physical activity interventions at this time may be effective in promoting or maintaining activity or reducing health inequalities. No evidence supported that transition to retirement period was or was not a significant point for intervention. Studies in older adults more generally indicated that a range of interventions might be effective for people around retirement age.
Loureiro, 2015 [47]	To identify programs that have been implemented with the goal of promoting the health of individuals and their families during their adjustment to retirement	Interpretive or critical studies	Individuals newly retired (within five years of retirement) and their families	I: Any C: Any	Any	N = 0 Authors could not identify any program.
Heaven, 2013 [20]	What kinds of intervention have been developed to promote social roles in retirement? How much have they improved perceived roles? Have these roles improved health or well-being?	Intervention studies	Participants were included if they fulfilled one of the following conditions 3: 1-Median age was between 55 and 70 years 2-Had been selected for the study because they were about to retire or had retired within the past two years; or 3-If they were selected for the study because they had a partner who met one of these criteria	I: Interventions that could extend or support the participants’ social roles C: Any	Participants’ perception of their social roles or well-being	N = 11 Social roles are linked to well-being outcomes through the way in which they are interpreted (such as providing feelings of worth, purpose, or perceptions of usefulness and status)
Wilson, 2007 [48]	To assess the state of science and accumulated evidence on of health promotion at retirement	Any	Retired population	I: Any C: Any	Any	N = 0 Authors did not identify any experimental trial.

RCT: Randomized Control Trial.

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
