# Peer review of "Interventions across the Retirement Transition for Improving Well-Being: A Scoping Review"

_ijerph, 2020, doi:10.3390/ijerph17124341_

Round 1
Reviewer 1 Report
The topic of the paper relates to interventions around work-to-retirement transition that aim to improve people's well-being around this time. The subject seems understudied and the authors chose scoping review as means to understand it better correctly. This said, I have several concerns and remarks, and I hope that Authors will benefit from these comments in the future versions of their manuscript.
- The introduction misses clear definitions of constructs that were used as search criteria. For example, it is only in the discussion that the reader learns that the Authors did in fact had a pre-existing notion of what they meant by well-being. The theoretical frames of such construct should be placed in the introduction.
- The introduction in shape and reasoning line is very similar to the one in BMJ paper, posing a question of self-plagiarism. The Authors should reconstruct the introduction more intensively.
- The aim of the scoping review is to gain a broad understanding and description of the field. I agree that the method chosen here is helpful in answering a question about what sort of interventions are applied during this transition period, as well as showcase what sort of settings and professionals are involved in it. However, I have doubts about research question 2 which involves assessing "impact and effectiveness of these interventions on the present and future well-being". I believe that this question requires a systematic review and/or meta-analysis, looking at effect sizes in RCTs, with short- and long-term effects of such interventions. I wonder, though, whether this research question could be answered with the method that the Authors chose.
- It is not clear whether there is any overlap between single studies reported in the first section of the manuscript and the systematic reviews summary. If the latter focused on the interventions around work-to-retirement time, it is only logical that they may have included the former. Thus, it should be mentioned whether or not this took place. If so, how does it affect conclusions? If not, why did the reviews miss these interventions (apart from the logical time order constraint)?
- The aim of turning to stakeholders and the how it was determined who stakeholders might be should be described in Method section. Are any important potential stakeholders missing? If so, this should be mentioned in the limitations.
- While the Authors mention that grey literature exists containing data on ageing and retirement, they limit themselves to describing what databases these are and what information they contain, without including any possible inferences or results. In fact, while the topic of the paper relates to the interventions, the data being revoked is about aging/retirement and health in general. In that case, I wonder whether this additional information actually brings anything to the table results-wise. At this point, I would say - not. This said, I would encourage authors to revoke this in the discussion commenting that there are more sources of information on this topic that should be reviewed/summarized by future researchers.
- Finally, given the aim of the scoping review, I would strongly encourage the Authors to identify more clearly what in the topic of work-retirement transition 1) is already known, 2) requires more knowledge (e.g., mixed results, low quality research), 3) seems completely understudied. Please, describe possible directions of future research in this important topic.
Thank you for the opportunity to read this manuscript!
Author Response
1. The introduction misses clear definitions of constructs that were used as search criteria. For example, it is only in the discussion that the reader learns that the Authors did in fact had a pre-existing notion of what they meant by well-being. The theoretical frames of such construct should be placed in the introduction.
RESPONSE: (See number 2)
2. The introduction in shape and reasoning line is very similar to the one in BMJ paper, posing a question of self-plagiarism. The Authors should reconstruct the introduction more intensively.
RESPONSE: Thank you for the suggestions. In agreement with your recommendations the introduction has been rewritten developing a different line of reasoning and including references to relevant constructs for this study.
3. The aim of the scoping review is to gain a broad understanding and description of the field. I agree that the method chosen here is helpful in answering a question about what sort of interventions are applied during this transition period, as well as showcase what sort of settings and professionals are involved in it. However, I have doubts about research question 2 which involves assessing "impact and effectiveness of these interventions on the present and future well-being". I believe that this question requires a systematic review and/or meta-analysis, looking at effect sizes in RCTs, with short- and long-term effects of such interventions. I wonder, though, whether this research question could be answered with the method that the Authors chose.
RESPONSE: We agree with the reviewer that, while assessing an intervention effectiveness would be better achieved with a systematic review, scoping reviews allow to identify and synthetize the types of available evidence in a given field – which was actually the purpose of our researchTherefore, we changed the question “What is the impact and effectiveness of these interventions on the present and future well-being of this population?” for “What is the current evidence of these interventions for improving well-being of this population?”. We appreciate the opportunity to reflect on this issue.
4. It is not clear whether there is any overlap between single studies reported in the first section of the manuscript and the systematic reviews summary. If the latter focused on the interventions around work-to-retirement time, it is only logical that they may have included the former. Thus, it should be mentioned whether or not this took place. If so, how does it affect conclusions? If not, why did the reviews miss these interventions (apart from the logical time order constraint)?
RESPONSE: This is indeed a question the whole team discussed a lot during the development of the scoping review. As R1 points out, since the research question is similar, there might be an overlap between experimental trials and systematic reviews. However, when checking this situation with the reference list of the systematic reviews we were able to see that not all trials that we had identified through our search strategy were included in the reviews, nor all the papers included in the systematic reviews had made it through our inclusion/exclusion criteria. Main factors explaining this was the different ways in which the retirement transition and health and wellbeing indicators had been conceptualized. Therefore, we chose to not plot together results from single trials and systematic reviews, but present them separately. This allows the reader to better compare the results of single trial studies and systematic reviews. This is especially important in this case since the inclusion criteria for “transition to retirement” of reviews was found to be very heterogeneous.
5.The aim of turning to stakeholders and the how it was determined who stakeholders might be should be described in Method section. Are any important potential stakeholders missing? If so, this should be mentioned in the limitations.
RESPONSE: Thank you for your suggestion. We have now addressed these ideas in the manuscript by including the reason why we decided to contact the different stakeholders. In addition, we have included in the limitations section a sentence regarding any missing stakeholders.
6. While the Authors mention that grey literature exists containing data on ageing and retirement, they limit themselves to describing what databases these are and what information they contain, without including any possible inferences or results. In fact, while the topic of the paper relates to the interventions, the data being revoked is about aging/retirement and health in general. In that case, I wonder whether this additional information actually brings anything to the table results-wise. At this point, I would say - not. This said, I would encourage authors to revoke this in the discussion commenting that there are more sources of information on this topic that should be reviewed/summarized by future researchers.
RESPONSE: Thank you for this point. We agree that the content of the grey literature section does not relate to the objective of this research which is focused on interventions. The grey literature search did not report any findings in regard to interventions within the work-to-retirement transition, therefore we intended to synthetize the available information on observational data. We have specified this idea in the grey literature section as well as in the discussion.
7. Finally, given the aim of the scoping review, I would strongly encourage the Authors to identify more clearly what in the topic of work-retirement transition 1) is already known, 2) requires more knowledge (e.g., mixed results, low quality research), 3) seems completely understudied. Please, describe possible directions of future research in this important topic.
RESPONSE: Thank you for the advice, the mentioned topics have been added to the discussion section.
Reviewer 2 Report
Dear authors, i thank you for research on the broad range of topics concerning aging. I would like to ask you to make a few changes in order to, i hope, to improve this paper. First I wold like for you to include another sy stematic review, and i am surprised you did not see it - Klugar et al, The personal active aging strategies of older adults in euorpe, publislhed with the Joanna briggs institute; second is some work by Roberson, D. N., Jr. especially in the area of personal learning, or self directed learning, in the introduction ? because so much of your work is about learning on ones own. Other specific concerns - 1. the title is too long, make it shorter - such as Learning how to REtire; 2. there should not be any initials in the abstract - (PPI). 3. Please explain what is a scoping review? 4. The use of ( ) is not a effective way to communicate in academic work. #59. 5. Drop self inflating words such as 'high sensitive' in #85. Grey list of literature is not really appropriate here in this review. 7. The conclusion is simply not clear - you have reviewed a lot of literature that has reviewed others, however, you have not made a specirfic statement which answers your purpose of studey or question of interest. You should be able to clearly see - "As a result of this research we can state there are four interventions participants have utilized in order to promote well being during retirement, the first is ............., second...............etc.
Author Response
1. the title is too long, make it shorter - such as Learning how to REtire;
RESPONSE: We appreciate the reviewer’s suggestion, however following the PRSIMA extension for scoping reviews, the title already includes the term scoping review as well as the key components that inform the eligibility criteria of the scoping review (population, concept, context).
2. there should not be any initials in the abstract - (PPI).
RESPONSE: This has been changed in the abstract section.
3. Please explain what is a scoping review?
RESPONSE: We appreciate your suggestion. In lines 80-83 we refer to cite the methodological framework for the scoping review followed for the development of this review.
4. The use of ( ) is not a effective way to communicate in academic work. #59.
RESPONSE: Thank for your comment, this has been edited in the new version of the manuscript.
5. Drop self inflating words such as 'high sensitive' in #85.
RESPONSE: Thank you for your comment. Developing a ‘highly sensitive search strategy’ is one of the most important steps when conducting a scoping or a systematic review, therefore the term “high sensitive” is commonly used in this type of method.
https://handbook-5-1.cochrane.org/chapter_6/6_4_11_1_the_cochrane_highly_sensitive_search_strategies_for.htm
6. Grey list of literature is not really appropriate here in this review.
RESPONSE: Thank you for this point. We agree that the content of the grey literature section does not relate to the objective of this research which is focused on interventions. They grey literature search didn’t report any findings in regard to interventions within the work-to-retirement transition, therefore we intended to synthetize the available information on observational data. We have specified this idea in the grey literature section as well as in the discussion section.
7. The conclusion is simply not clear - you have reviewed a lot of literature that has reviewed others, however, you have not made a specirfic statement which answers your purpose of studey or question of interest. You should be able to clearly see - "As a result of this research we can state there are four interventions participants have utilized in order to promote well being during retirement, the first is ............., second...............etc.
RESPONSE: We appreciate your suggestion, the conclusion has been modified and made clearer.
Round 2
Reviewer 1 Report
Dear Authors, thank you for addressing the comments stipulated with regards to this article. I have one final comment.
In the conclusions, the Authors state that "as a result of this scoping review, we can state that more research on how to design and implement interventions to promote overall wellbeing during the work-to-retirement transition is needed". Implementation relates to the process of how the intervention is introduced to the workplace/participants. While scoping for settings or personnel involved partially answers this question, other aspects of implementation should be included to make this conclusion (e.g., comment on the intensity of the intervention, participant involvement in the design, whether there was ongoing evaluation of the process, etc.). Thus, I think that if authors want to say that more research is needed on the implementation, they should assess the studies they reviewed with regards to these aspects and summarize what is known so far. An answer to this question could provide knowledge on HOW such interventions are introduced and implemented. Otherwise, this conclusion should be toned down to the main findings, and the idea on evaluating the aspects of implementation of such interventions could be presented as future directions of research.